# Sphingolipid Inhibitors as an Alternative to Treat Candidiasis Caused by Fluconazole-Resistant Strains

**DOI:** 10.3390/pathogens10070856

**Published:** 2021-07-07

**Authors:** Rodrigo Rollin-Pinheiro, Brayan Bayona-Pacheco, Levy Tenorio Sousa Domingos, Jose Alexandre da Rocha Curvelo, Gabriellen Menezes Migliani de Castro, Eliana Barreto-Bergter, Antonio Ferreira-Pereira

**Affiliations:** 1Laboratório de Química Biológica de Microrganismos, Departamento de Microbiologia Geral, Instituto de Microbiologia Paulo de Góes, Universidade Federal do Rio de Janeiro (UFRJ), Rio de Janeiro 21941-902, Brazil; rodrigorollin@gmail.com (R.R.-P.); eliana.bergter@micro.ufrj.br (E.B.-B.); 2Laboratório de Bioquímica Microbiana, Departamento de Microbiologia Geral, Instituto de Microbiologia Paulo de Góes, Universidade Federal do Rio de Janeiro (UFRJ), Rio de Janeiro 21941-902, Brazil; bbayona@uninorte.edu.co (B.B.-P.); levydomingos@yahoo.com.br (L.T.S.D.); alexandrecurvelo@hotmail.com (J.A.d.R.C.); g.migliani@gmail.com (G.M.M.d.C.); 3Departamento de Medicina, División Ciencias de la Salud, Universidad del Norte, Km 5, vía Puerto Colombia, Área Metropolitana de Barranquilla 081007, Colombia

**Keywords:** *Candida*, sphingolipids, myriocin, fungal infections

## Abstract

*Candida* species are fungal pathogens known to cause a wide spectrum of diseases, and *Candida albicans* and *Candida glabrata* are the most common associated with invasive infections. A concerning aspect of invasive candidiasis is the emergence of resistant isolates, especially those highly resistant to fluconazole, the first choice of treatment for these infections. Fungal sphingolipids have been considered a potential target for new therapeutic approaches and some inhibitors have already been tested against pathogenic fungi. The present study therefore aimed to evaluate the action of two sphingolipid synthesis inhibitors, aureobasidin A and myriocin, against different *C. albicans* and *C. glabrata* strains, including clinical isolates resistant to fluconazole. Susceptibility tests of aureobasidin A and myriocin were performed using CLSI protocols, and their interaction with fluconazole was evaluated by a checkerboard protocol. All *Candida* strains tested were sensitive to both inhibitors. Regarding the evaluation of drug interaction, both aureobasidin A and myriocin were synergic with fluconazole, demonstrating that sphingolipid synthesis inhibition could enhance the effect of fluconazole. Thus, these results suggest that sphingolipid inhibitors in conjunction with fluconazole could be useful for treating candidiasis cases, especially those caused by fluconazole resistant isolates.

## 1. Introduction

*Candida* species cause a wide spectrum of infections in humans, ranging from superficial mycosis, especially associated to skin and vaginal mucosae, to life-threatening disseminated candidiasis [1]. *Candida albicans* and *Candida glabrata* are the most frequent species associated to invasive infections, being responsible for about 46% and 24%, respectively [2]. Candidiasis is commonly associated with different pathologies, such as HIV infection, organ transplantation, cancer and diabetes, which contribute to a mortality rate of up to 60% [3]. In addition, it has been considered the most frequent fungal disease associated to healthcare units and the fourth most prevalent nosocomial infection [4].

The current antifungal drugs available to be used against fungal infections are limited to only four classes: azoles, which block the enzyme lanosterol 14-α demethylase and, thus, disrupt ergosterol synthesis; polyenes, which directly bind to ergosterol found in the plasma membrane and cause the release of cytoplasmic content; echinocandins, which inhibit the enzyme β(1,3)-glucan synthase and affect cell wall synthesis; and fluoropyrimidine analogs, which block DNA synthesis [5]. The first choice of drug to treat candidiasis is fluconazole, an azole antifungal drug, but resistant strains have been emerging over the last decades, causing a significant impact in public health [6,7]. Multi-resistant strains have been isolated, which express different types of transporters or display over-expressed azole targets [8]. In addition, alternative therapeutic options are limited due to low diversity of antifungal classes and high level of toxicity and side effects [9].

For these reasons, there is an urgent need to find alternative therapeutic approaches to obtain better results in treating patients who carry resistant strains. Different fungal cell components have been studied as potent new targets for the development of antifungal drugs. In this context, sphingolipids have been considered interesting candidates. Several studies have shown that sphingolipids, mainly glucosylceramide, play crucial roles in fungal growth, cellular signaling and virulence [10,11,12]. In *C. albicans*, *Cryptococcus neoformans*, *Penicillium digitatum*, and *Aspergillus fumigatus*, mutants which do not express the glucosylceramide synthase gene displayed alterations in plasma membrane, growth, and virulence in infection models [13,14,15,16,17]. In addition, some compounds that inhibit sphingolipid biosynthesis, such as aureobasidin A and myriocin, have been shown to present antifungal activity with low minimum inhibitory concentration (MIC) in a variety of fungal pathogens, including *Candida* and *Aspergillus* species [18,19,20,21,22]. More recently, a class of drugs called acylhydrazones was described which affect the synthesis of glucosylceramides of *C. neoformans*, *C. albicans*, *A. fumigatus*, and *Pneumocystis murina*, but not those from mammalian cells [23].

Due to the increasing resistance of *Candida* species associated with invasive infections and the potential of targeting sphingolipids, the present study aimed to test two inhibitors of sphingolipid synthesis, aureobasidin A and myriocin. Both drugs were evaluated against different *Candida* clinical isolates, including some that were described as strains that overexpress transmembrane transporters (ABC and MFS) related to multidrug resistance phenotype [24,25]. Aureobasidin A and myriocin inhibit inositolphosphorylceramide (IPC) synthase and glucosylceramide (GlcCer) synthase, respectively, which are two key enzymes for the synthesis of the most important sphingolipids found in fungi such as IPC and GlcCer.

## 2. Results

### 2.1. Antifungal Effect against Candida Strains

Aureobasidin A and myriocin were initially tested against all *Candida* strains used in the study (Table 1). ATCC strains were used as a control because they do not to express resistance mechanisms. On the other hand, clinical isolates are highly resistant to fluconazole (MIC > 256 µg/mL) and present different resistance mechanisms as pointed out in Materials and Methods section.

Aureobasidin A inhibits fungal growth at 0.5 µg/mL for ATCC strains (*C. albicans* and *C. glabrata*) and at 0.25 µg/mL for clinical isolates (109, 1114, and 1299) (Table 2). Myriocin presents antifungal activity at 2.0 and 1.0 µg/mL for ATCC *C. albicans* and *C. glabrata*, respectively, and at 0.5, 1.0, and 0.25 µg/mL for clinical isolates 109, 1114, and 1299, respectively (Table 2).

Cell viability decreased at MIC values, as evaluated by XTT-reduction assay, indicating that both inhibitors present fungicidal effect (Figure 1).

### 2.2. Interaction between Aureobasidin A, Myriocin, and Fluconazole

As pointed out previously, fluconazole is the first choice in treating candidiasis and the emergence of fluconazole-resistant strains is a concern in healthcare settings, because it is related to high mortality of infected patients. For this reason, a synergic effect of sphingolipid inhibitors and fluconazole could be useful in order to improve treatment of patients infected with resistant *Candida* species.

Both sphingolipid inhibitors display a synergic effect with fluconazole (Table 3), except aureobasidin A in strain 12-99. Aureobasidin A reduced fluconazole MIC from more than 256 µg/mL (strains 109 and 12-99) or 128 µg/mL (strain 1114) to 32, 16, or 128 µg/mL for strains 109, 1114, and 12-99, respectively. FICI values were 0.1874, 0.25, and 0.56 for strains 109, 1114, and 12-99, respectively, indicating that a synergic effect occurs between aureobasidin A and fluconazole for most of the strains used.

On the other hand, myriocin decreased fluconazole MIC to 64 (strain 109), 32 (strain 1114) and 16 µg/mL (strain 12-99) (Table 3). Corresponding FICI values were 0.375, 0.5 and 0.31 for strains 109, 1114, and 1299, respectively, confirming a synergic effect between myriocin and fluconazole.

A graphical representation of synergism data is shown in Figure 2.

### 2.3. Cytotoxicity of Aureobasidin A and Myriocin

In order to test if aureobasidin A and myriocin are toxic to mammalian cells at the same concentrations found effective in previous experiments, a cytotoxicity assay was performed on murine macrophages (RAW 264.7). Compared to untreated cells, the control of cells treated with 1% dimethylsulfoxide (DMSO) presented 80% viability whereas myriocin treatment led to 60% viability, demonstrating a decrease of approximately 20% compared to DMSO-treated cells (Figure 3). Regarding aureobasidin A treatment, RAW cell viability was not affected up to 2.5 µg/mL, which is more than 10-fold higher than the concentration presenting synergism with fluconazole.

These data suggest that aureobasidin A is not toxic at concentrations used in this work, whereas myriocin is partially toxic.

### 2.4. Effect of Aureobasidin A and Myriocin on the Lifespan of Wild Type Caenorhabditis elegans

To check the toxicity of compounds now against living organisms, we did a survival test using a wild-type of *C. elegans* worm. The worms were tested in the presence of 0.5 µg/mL of both compounds and in the presence of 0.1% DMSO as a control. After 4 days of analysis regarding the survival of the worms in the presence of the compounds, it can be observed that only myrocin at 0.5 µg/mL was toxic to the worms since in the case of aureobasidin A, at the same concentration, the survival was practically the same (approximately 98%) when compared to the control (Figure 4). These results corroborate what was observed in the cytotoxicity assay using macrophages (Figure 3), where only myrocin seemed to be more toxic than aureobasidin A.

## 3. Discussion

Infections caused by *Candida* species represent a significant concern in clinical settings due to their high morbidity and mortality, as well as the emergence of resistant isolates [27,28]. Thus, the study of new alternatives to treat candidiasis, especially those caused by resistant strains, is an urgent need.

Sphingolipids are a potential new target for drug development. They are considered a good candidate because fungal sphingolipids are structurally different from the human counterparts and due to their crucial roles in fungal growth, cellular signaling and pathogenesis also. Several studies demonstrated that different compounds are able to block different steps of sphingolipid biosynthesis and therefore present antifungal activity. These compounds—such as myriocin, fumonisin B1, aureobasidin A, and D-*threo*-1-phenyl-2-decanoylamino-3-morpholino-1-propanol (D-threo-PDMP)—act by blocking serine palmitoyltransferase, ceramide synthases, IPC synthase, and GlcCer synthase, respectively [12].

The present study aimed to use two of these compounds, aureobasidin A and myriocin, in order to evaluate their activity against *Candida* species. *C. albicans* ATCC (10231D-5) and *C. glabrata* ATCC (2001D-5) were used, as well as three clinical isolates highly resistant to fluconazole, *C. glabrata* strain 109 (over-expression of the CDR1 gene), *C. albicans* strain 1114 (over-expression of the MDR1 gene) and *C. albicans* strain 12-99 (over-expression of the genes ERG11, CDR1, CDR2, MDR1) (Table 1).

Aureobasidin A displayed MICs of 0.5 µg/mL for ATCC strains and 0.25 µg/mL for clinical isolates, whereas myriocin presented MICs ranging from 0.25 µg/mL to 2.0 µg/mL (Table 2, Figure 1). These data are in accordance with the literature, since it has also been demonstrated that aureobasidin A at concentrations of 2.0–3.5 µg/mL inhibited ATCC strains of *C. albicans*, *C. glabrata*, *C. tropicalis*, *C. parapsilosis*, and *C. krusei* [29]. Clinical isolates were also evaluated by Tan and Tay [21], who showed MICs for aureobasidin A of 4 µg/mL for *C. albicans* and 1 µg/mL for non-*albicans* isolates. This inhibitory effect has already been described to occur due to ceramide intoxication and deprivation of essential IPCs [19]. In addition, aureobasidin A has already been demonstrated to inhibit IPC synthase activity even at nanomolar levels, suggesting that its antifungal action might be a result of alterations on the biosynthesis of sphingolipids [29]. Kumar and colleagues have also shown that in vitro treatment of *Candida auris* with aureobasidin A leads to a deregulation of many intermediates of sphingolipid biosynthetic pathway [30].

Regarding myriocin, it presented MICs ranging from 0.25 µg/mL to 2.0 µg/mL (Table 2, Figure 1). De Melo and colleagues reported a similar MIC value of 0.12 µg/mL for *C. albicans* SC5314 [31]. Recently, myriocin was tested against some *Candida* strains, including isolates resistant to voriconazole, and MICs were found varying between 0.125–4.0 µg/mL [32]. Aureobasidin A and myriocin also affect *Candida* biofilms, and it was due to modification in lipid composition and to the alteration in lipid raft organization and plasma membrane [20,32]. The effect of myriocin on *Candida* cells has been recently evaluated by Yang and colleagues, who demonstrated that a disruption of plasma membrane is observed when different species are treated with myriocin [32]. Similar data have also been shown in other pathogenic fungi, such as *Scedosporium boydii*, in which myriocin led to alterations on plasma membrane resulting in higher susceptibility to membrane stressors such as SDS [33]. In *Aspergillus fumigatus*, myriocin treatment led to a decrease in phytoceramide content, suggesting that this inhibitor also alter the regulation of sphingolipid production [22]. Thus, the effect of myriocin and aureobasidin A against different pathogenic fungi suggests that the disruption of sphingolipid biosynthesis seems to display a conserved antifungal activity, although more studies are needed using other samples and compounds [18,20,22,34,35].

Synergistic effect of two different drugs is a promising alternative to enhance efficacy of the current antifungals. This approach uses two known drugs combined, which are already approved to be used in clinical settings, and their toxicity was already determined. This is a great advantage when compared to the costly and time-consuming development of new drugs [36]. The best-known example of synergism is the combination of fluorocytosine and amphotericin B, which is the gold standard treatment for cryptococcosis [36,37]. However, very few studies describe synergistic effects of sphingolipid inhibitors and the current antifungal drugs. De Melo and colleagues demonstrated that myriocin is synergistic to amphotericin B but not to fluconazole [31]. However, only one susceptible strain and no fluconazole-resistant isolate was used, so more studies are needed in this field. Since fluconazole is the first choice to treat *Candida* invasive infections with high mortality and resistant strains have been emerging in recent years, it is crucial to develop treatment alternatives. Our data showed that both aureobasidin A and myriocin present synergistic effects with fluconazole on almost all clinical isolates tested in this work (Table 3, Figure 2). As mentioned, the clinical isolates used are highly resistant to fluconazole, which suggests that synergy is a promising option to be used in patients carrying fluconazole-resistant yeast strains. Myriocin also presents synergism with fluconazole in *Scedosporium boydii*, a pathogenic filamentous fungus, suggesting that this effect could be conserved among other pathogens [33], suggesting that targeting fungal sphingolipids in combination with azoles is promising in order to treat fungal infections, especially in cases where resistance to azoles lead to a failure in treatment success.

A key point and concern of using sphingolipid inhibitors to treat fungal infections is their cytotoxic effect in humans. For instance, fumonisin B is a mycotoxin produced by *Fusarium* species that also display toxicity to mammalian cells [38,39]. In the present study, cytotoxic assays showed that myriocin is partially toxic in RAW cells, whereas aureobasidin A is not toxic (Figure 3), and the same profile of toxicity was observed when both compounds were tested against live *C. elegans* (Figure 4), suggesting that both drugs (specially aureobasidin A) could be considered in addition to fluconazole. Considering that fluconazole resistance in *Candida* isolates is a serious problem in clinical healthcare settings, sphingolipid inhibitors were shown to be potential therapeutic options and more studies are needed to explore their use as an alternative approach when administered in combination with fluconazole.

## 4. Conclusions

The present work showed the two sphingolipid inhibitors, myriocin and aureobasidin A, display antifungal activity against *C. albicans* and *C. glabrata*, not only against ATCC strains but also clinical isolates, suggesting that these compounds are active against strains presenting resistance mechanisms to the current antifungal drugs used in clinical settings. Myriocin and aureobasidin A also presented synergistic interaction with fluconazole, indicating that they could be a promising approach as a combined therapy especially to treat infections caused by resistant strains.

Toxicity analyses revealed that both drugs do not display significant toxic effect, especially in the *C. elegans* model, which reinforces their potential as an alternative therapy when combined with fluconazole. Further studies are needed to evaluate in vivo activity of this approach and to clarify the promising use of sphingolipid inhibitors as alternatives to treat *Candida* infections.

## 5. Materials and Methods

### 5.1. Cell Lineages and Reagents

A total of five strains were used in this study (Table 1). *C. albicans* ATCC 10231D-5 and *C. glabrata* ATCC 2001D-5 were used as standard. Three clinical isolates displaying resistance patterns to fluconazole were also evaluated, *C. glabrata* 109 strain (which displays overexpression of *CDR1* gene that encode a ABC transporter), *C. albicans* 1114 strain (which displays overexpression of *MDR1* gene that encode a MFS transporter) and *C. albicans* 12-99 strain (which displays *ERG11*, that confers resistance by mutation or overexpression of 14-α demethylase involved in ergosterol synthesis; *CDR1*, *CDR2* and *MDR1* genes that encode ABC and MFS transporters), kindly provided by Theodore White from University of Missouri, USA. For all experiments, the strains were grown on Yeast Extract Peptone Dextrose (YPD) agar and transferred to YPD broth and incubated at 37 °C for 18 h under agitation.

Cytotoxicity assays were carried out using the murine macrophage-derived cell lines RAW 264.7.

Aureobasidin A (Sigma–Aldrich, St. Louis, MO, USA), myriocin (Sigma–Aldrich, St. Louis, MO, USA) and fluconazole (University pharmacy, UFJF, Juiz de Fora-MG, Brazil) were used in susceptibility and synergism tests.

### 5.2. Susceptibility Tests with Aureobasidin A and Myriocin and Interaction with Fluconazole

The susceptibility assay was performed to determine the minimal inhibitory concentration (MIC) of aureobasidin A and myriocin, according to Clinical Laboratory Standards Institute (CLSI) M60 protocol. Both compounds were used in a concentration range of 0.031–4.0 µg/mL and MIC_90_ was determined when fungal growth presented 90% of inhibition compared to a positive control of untreated cells. Briefly, yeasts were inoculated in sterile 96-well plates in 200 μL of RPMI medium (Roswell Park Memorial Institute), so that they reached the concentration of 5 × 10^3^ cells/mL in the presence of 1:2 dilutions of each compound. The 96-well plates were incubated at 37 °C for 48 h with shaking (100 rpm). Cell growth was evaluated using a microplate reader (iMark, Bio-Rad, Hercules, CA, USA) at 600 nm.

*Candida* cell viability was evaluated after MIC determination by using the XTT-reduction technique, according to Rollin-Pinheiro and colleagues [33]. Briefly, after fungal growth as mentioned above, a 0.5 mg/mL XTT solution in PBS was added to the 96-well plates and cells were incubated at 37 °C for 2 h protected from light. Further, optical density was measured using a spectrophotometer (SpectraMax^®^ i3x, Molecular Devices^®^, San José, CA, EUA) at 490 nm to evaluate cell viability.

Interaction analysis of aureobasidin A and myriocin with fluconazole was performed using the checkerboard method according to Reis de Sá and colleagues [40]. Aureobasidin A and myriocin concentrations ranged from 0.0078–2.0 µg/mL and fluconazole concentration from 16–256 µg/mL. After 48 h of growth at 37 °C under agitation, the fractional inhibitory index (FICI) was calculated according to the formula (MIC combined/MIC drug A alone) + (MIC combined/MIC drug B alone), where A is aureobasidin A or myriocin and drug B is fluconazole. Interaction was classified according to the following parameter: ≤0.5, synergistic interaction; >0.5 to ≤4, no interaction; >4, antagonistic effect [41].

### 5.3. Cytotoxicity Assay

Cytotoxicity was analyzed by neutral red (NR) assay with modifications [42]. RAW 264.7 cell monolayer was harvested with a cell scraper and viable cells were counted using the Trypan blue exclusion method. 2 × 10^5^ macrophages per well were seeded in 96-well plates containing Dulbecco’s modified Eagle medium (DMEM) with 10% FBS and incubated in a controlled atmosphere of 5% CO_2_ at 37 °C for adhesion. Compounds were serial diluted in DMEM and cells were incubated at concentrations of 0.313, 0.625, 1.25, 2.50, 5, and 10 µg/mL at 37 °C, 5% CO_2_ for 48h. Cells without compounds were used as control. Absorbance was determined in a spectrophotometer at 595 nm (SpectraMax^®^ i3x, Molecular Devices^®^, San José, CA, EUA). Each test was performed in triplicate.

### 5.4. Caenorhabditis elegans Lifespan Assay

*C. elegans* strain N2 (wild isolate) was obtained from the Caenorhabditis Genetics Center at University of Minnesota (USA) and handled according to standard method [43]. Worms were maintained at 15 °C on nematode growth medium (NGM) and routinely maintained on *Escherichia coli* OP50 strain used as a normal diet for nematodes. Lifespan worm assay was performed as previously described [44,45] with small modifications. Briefly, synchronization of worms was achieved by preparing eggs from gravid adults using a solution containing NaOCl 6% and NaOH 5M; released eggs were washed with M9 buffer and allowed to hatch overnight in NGM agar plates. Synchronized young worms were collected by washing with M9 buffer. Approximately 20 worms were added to each well of 96-well plates containing 100 mL MB medium in the absence or presence of 0.5 µg/mL aureobasidin or 0.5 µg/mL myriocin. Then, the plates were incubated at 25 °C during 4 days without shaking and scored as live and dead-on daily basis. The survival ratio was calculated from the percentage of living worms out of total number of worms including living and dead animals. This experiment was independently conducted in two different days with a twofold analysis in each one.

### 5.5. Statistical Analyses

All experiments were performed in triplicate, in three independent experimental sets. Statistical analyses were performed using GraphPad Prism version 5.00 for Windows (GraphPad Software, San Diego, CA, USA). One-way analysis of variance using a Kruskal–Wallis nonparametric test was used to compare the differences between groups, and individual comparisons of groups were performed using a Bonferroni posttest. The 90–95% confidence interval was determined in all experiments.

## Figures and Tables

**Figure 1 pathogens-10-00856-f001:**
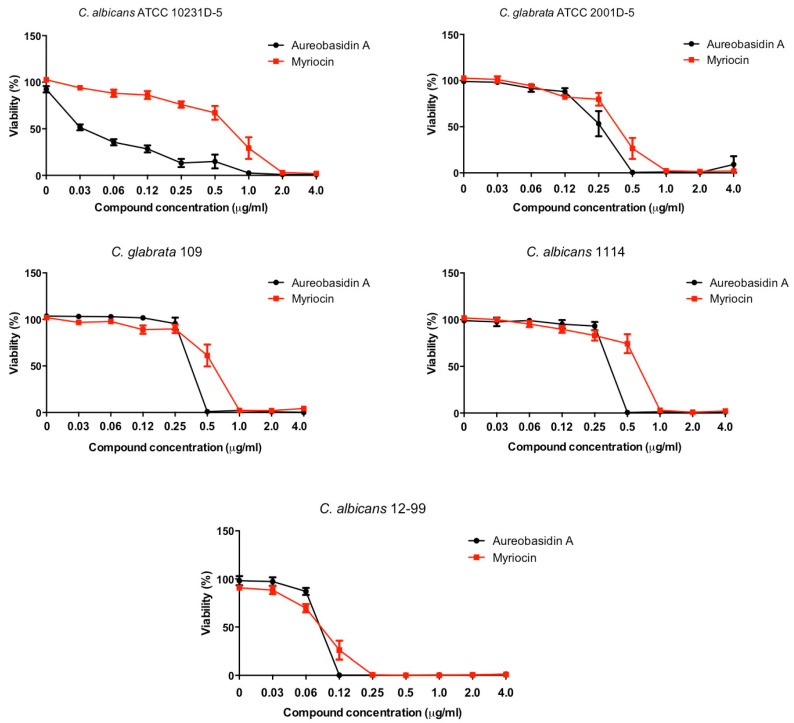
Viability of *Candida* ATCC (*C. albicans* ATCC 10231D-5 and *C. glabrata* ATCC 2001D-5) and clinical strains (*C. albicans* 1114 and 12-99, *C. glabrata* 109) in the presence of aureobasidin A and myriocin. Cells were grown in microplates containing RPMI at 37 °C for 48 h in the absence (0 µg/mL) or in the presence of aureobasidin A and myriocin. After the incubation time, viability was evaluated using the XTT-reduction assay. Cell viability was quantified using a microplate reader (Bio-Rad, Hercules, CA, USA) at 490 nm. Percentage of fungal growth was calculated considering the control (absence of all drugs) as 100%. Errors bars represent standard errors of means of different experiments in different days (*n* = 3).

**Figure 2 pathogens-10-00856-f002:**
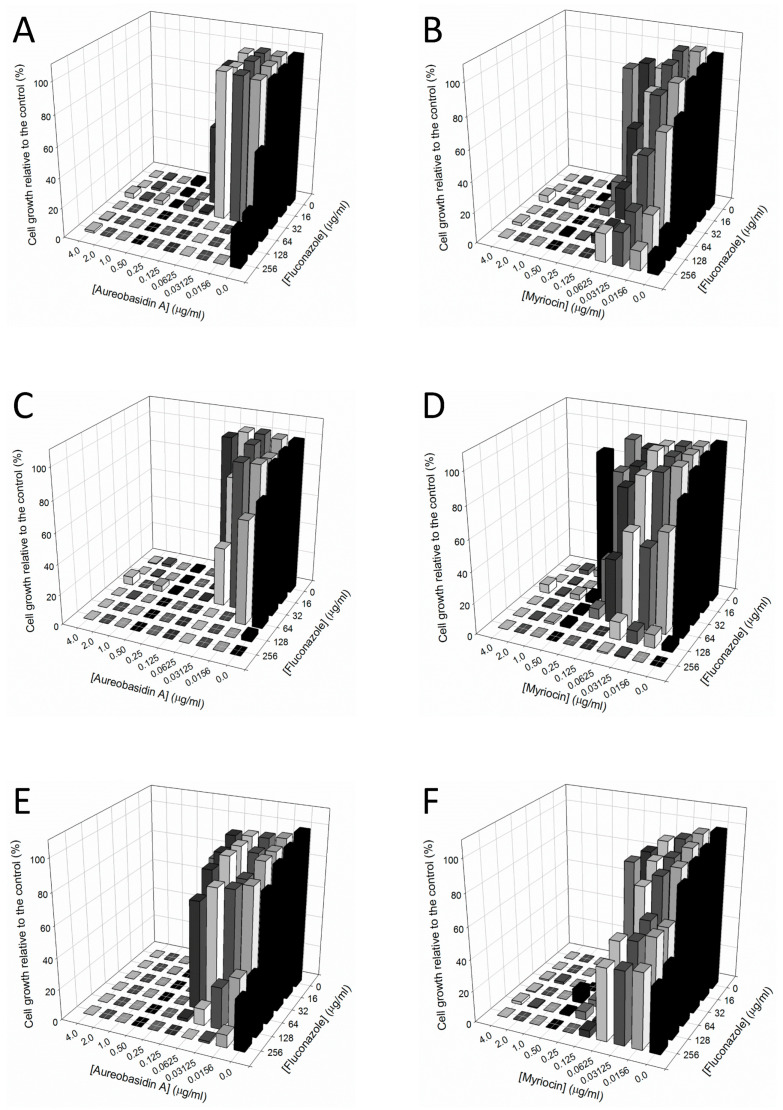
Evaluation of the interaction between aureobasidin A (**A**,**C**,**E**) and myriocin (**B**,**D**,**F**) with fluconazole. Analysis was performed with the isolates 109 (**A**,**B**), 1114 (**C**,**D**), and 1299 (**E**,**F**) and the results are shown using the software Sigma Plot 12.0. Cells were grown in microplates containing RPMI at 37 °C for 48 h in the absence (0 µg/mL) or in the presence of different combinations of aureobasidin A or myriocin with fluconazole. After the incubation time, cell growth was evaluated using a microplate reader (Bio-Rad, Hercules, CA, USA) at 600 nm. Percentage of fungal growth was calculated considering the control (absence of all drugs) as 100%.

**Figure 3 pathogens-10-00856-f003:**
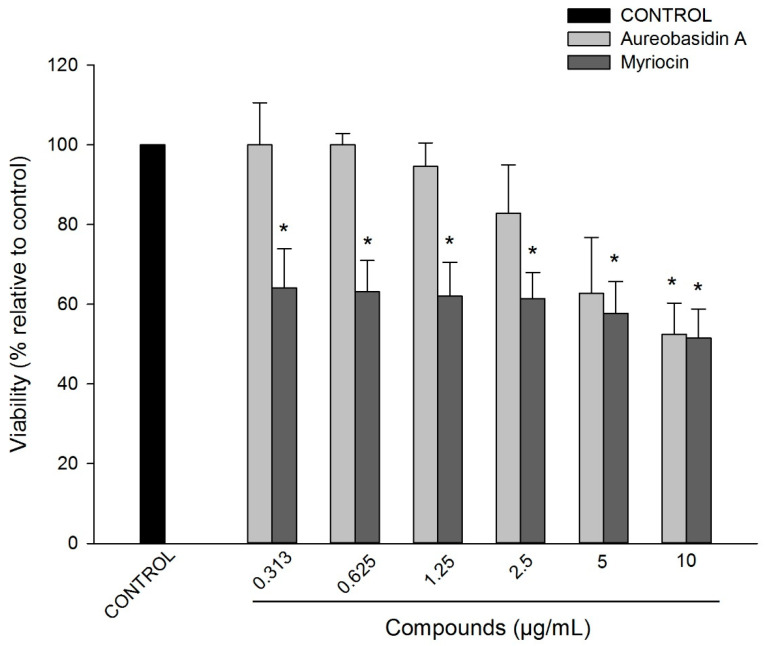
Cytotoxic assay of aureobasidin A and myriocin against murine macrophage (RAW 264.7) cell lineage for 48 h. The data represent the means of three independent experiments and the error bars represent the standard error. Solid black bar represents the control in the absence of compounds. * *p* < 0.05.

**Figure 4 pathogens-10-00856-f004:**
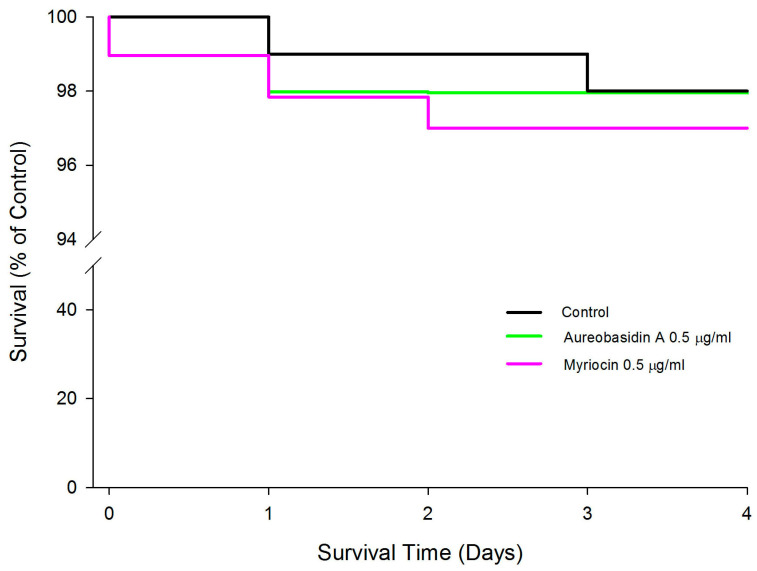
Effect of aureobasidin A and myriocin exposure on lifespan of *C. elegans* wild-type (N2) during five days. The data represent the means of two independent experiments that were collected daily. The control was performed in the absence of compounds but in the presence of same amount of DMSO (0.1%) present at tested compounds.

**Table 1 pathogens-10-00856-t001:** *Candida* strains used in the study.

Strains	Resistance Pattern	Reference
*C. albicans* (ATCC 10231D-5)	No resistance described	American Type Culture Collection *
*C. glabrata* (ATCC 2001D-5)	No resistance described	American Type Culture Collection *
*C. glabrata* (109)	*CDR1* gene overexpressed (ABC transporter)	[24]
*C. albicans* (1114)	*MDR1* gene overexpressed (MFS transporter)	[25]
*C. albicans* (12-99)	*ERG11*, *CDR1*, *CDR2* and *MDR1* genes overexpressed (ABC and MFS transporters)	[26]

* https://www.atcc.org/, (accessed on 30 June 2021).

**Table 2 pathogens-10-00856-t002:** *Candida* strains susceptibility to Aureobasidin A, Myriocin, and Fluconazole.

	* MIC_90_ (µg/mL)		
	Aureobasidin A	Myriocin	Fluconazole
*C. albicans* (ATCC 10231D-5)	0.5	2.0	<8
*C. glabrata* (ATCC 2001D-5)	0.5	1.0	<8
*C. glabrata* (109)	0.25	0.5	>256
*C. albicans* (1114)	0.25	1.0	>256
*C. albicans* (12-99)	0.25	0.25	>256

* MIC: minimal inhibitory concentration.

**Table 3 pathogens-10-00856-t003:** Interaction of aureobasidin A or myriocin with fluconazole tested in Candida clinical isolates presenting resistance to fluconazole.

	*Candida* Strains		
	109	1114	12-99
	MIC_90_ alone (µg/mL)		
Fluconazole	>256	128	>256
Aureobasidin A	0.25	0.25	0.25
Myriocin	0.5	1.0	0.25
	MIC_90_ combined (µg/mL)		
Aureo/Fluco	0.0156/32	0.03125/16	0.015/128
Myr/Fluco	0.0625/64	0.25/32	0.0625/16
	FICI		
Aureo/Fluco	0.1874 (synergic)	0.25 (synergic)	0.56 (no effect)
Myr/Fluco	0.375 (synergic)	0.5 (synergic)	0.31 (synergic)

Aureo: aureobasidin A; Fluco: fluconazole; Myr: myriocin; MIC: minimal inhibitory concentration; FICI: fractional inhibitory index.

## Data Availability

Not applicable.

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
