# Peer review of "Sphingolipid Inhibitors as an Alternative to Treat Candidiasis Caused by Fluconazole-Resistant Strains"

_pathogens, 2021, doi:10.3390/pathogens10070856_

Round 1
Reviewer 1 Report
The authors report that sphingolipid inhibitors in conjunction with fluconazole could be useful for treating candidiasis cases, especially those caused by fluconazole resistant isolates.
The paper is well written but I have some comments:
- The discussion is long and complex
- add the conclusions section
Author Response
REVIEWER 1:
The discussion is long and complex
ANSWER: The discussion was revised and some modifications were done to make it more clear and objective.
add the conclusions section
ANSWER: We did not put this topic before because it was not present at jounal template, but now, we accepted the suggestion and a conclusion topic was added to the final text.
Reviewer 2 Report
Dear authors,
This manuscript describes the important issue of pathogenic fungi becoming resistant to conventional anti-fungal drugs and the urgent need to find new, effective treatments. The paper is well-written and understandable, and the experimental details and results are appropriate for this study. I recommend the following comments and changes to improve the manuscript:
Minor errors:
Line 45: change ‘equinocandins’ to ‘echinocandins’.
Line 93: remove extra comma
Line 102, 111, 156: change ‘(0)’ to ‘(0 μg/ml)’
Line 182: Remove abbreviation of C. elegans
Line 186: DMSO mentioned in previous paragraph. Include abbreviation when first mentioned.
Line 189, 192: Change ‘aureobasidin’ to ‘aureobasidin A’
Line 190: Rephrase the last sentence of this paragraph
Line 197: ‘experiment’ to ‘experiments’
Line 216: the D referring to the chirality should be one font size smaller. Also, ‘threo’ should be italicised.
Line 278: change ‘produces’ to ‘produced’
Line 280: change ‘assay’ to ‘assays’
Line 281: change ‘same’ to ‘the same’
Line 294: remove ‘)’
Line 306: change ‘MIC90’ to ‘MIC90’
Line 321: reference is not in correct format
Line 323: change ‘2 x 105’ to ‘2 x 105’
Line 325, 327: change ‘CO2’ to ‘CO2’
Remove lines 331-338, since previous paragraph is repeated
Line 340: change ‘Caenorhabditis elegans (C. elegans)’ to ‘C. elegans’
Line 367: change/remove ‘X.X.’
Other questions/comments:
- In Table 1, the overexpression of the genes indicated encode for what? I think you should include this in the table or in the discussion. Eg. MDR1 encodes for MFS transporters. This will clarify the choice of Candida
- Line 84: It is said that ‘clinical isolates are highly resistant to fluconazole (MIC > 256 μg/ml) and present different resistance mechanisms as pointed out in Material and Methods section.’ These resistance mechanisms are not pointed out in the Material and Methods section.
- In Table 2, is it possible to give the exact MIC values for Fluconazole for the control strains (instead of reporting <8)?
- Is it possible to combine Figures 1 & 2 since the captions are nearly identical?
- In Table 3, label the first line as ‘Candida strains’. The middle half of this table is difficult to interpret at first. Possibly change to something like follows:
|
|
|
MIC combined (μg/ml) |
|
|
Aureo/Fluco |
0.0156/32 |
0.03125/16 |
0.015/128 |
|
Myr/Fluco |
0.0624/64 |
0.25/32 |
0.0625/16 |
Please check the FICI values reports in table 3. When using the equation outlined in the Material and Methods section, different values are obtained. In particular, for Candida strain 12-99 the FICI value for Aureo/Fluco should be greater than 0.5 indicating no synergic interaction.
- Line 167: Include the amount/percentage of DMSO present in the control.
- Line 173: Myriocin is found to be partially toxic. How will this effect it’s potential as a therapeutic agent for Candida infections?
- Figure 5: Please include the line for the control from 0-1 day. It is unclear whether it is at 99 or 100 %.
- Figure 5: I suggest using a different colour line for the control in the presence of 5% DMSO in the inset. This will clearly differentiate between the two controls in this figure.
- Lines 210-212: This reference and example is discussed again in lines 272-276. It is not necessary to mention twice in the discussion.
- Line 309-310: The XTT reduction assay is mentioned here. It would be good to include a brief summary of the procedure you used.
Author Response
REVIEWER 2:
Line 45: change ‘equinocandins’ to ‘echinocandins’.
ANSWER: It was corrected in the text.
Line 93: remove extra comma
ANSWER: It was corrected in the text.
Line 102, 111, 156: change ‘(0)’ to ‘(0 μg/ml)’
ANSWER: It was corrected in the text.
Line 182: Remove abbreviation of C. elegans
ANSWER: It was corrected in the text.
Line 186: DMSO mentioned in previous paragraph. Include abbreviation when first mentioned.
ANSWER: It was corrected in the text.
Line 189, 192: Change ‘aureobasidin’ to ‘aureobasidin A’
ANSWER: It was corrected in the text.
Line 190: Rephrase the last sentence of this paragraph
ANSWER: It was corrected in the text.
Line 197: ‘experiment’ to ‘experiments’
ANSWER: It was corrected in the text.
Line 216: the D referring to the chirality should be one font size smaller. Also, ‘threo’ should be italicised.
ANSWER: It was corrected in the text.
Line 278: change ‘produces’ to ‘produced’
ANSWER: It was corrected in the text.
Line 280: change ‘assay’ to ‘assays’
ANSWER: It was corrected in the text.
Line 281: change ‘same’ to ‘the same’
ANSWER: It was corrected in the text.
Line 294: remove ‘)’
ANSWER: It was corrected in the text.
Line 306: change ‘MIC90’ to ‘MIC90’
ANSWER: It was corrected in the text.
Line 321: reference is not in correct format
ANSWER: It was corrected in the text.
Line 323: change ‘2 x 105’ to ‘2 x 105’
ANSWER: It was corrected in the text.
Line 325, 327: change ‘CO2’ to ‘CO2’
ANSWER: It was corrected in the text.
Remove lines 331-338, since previous paragraph is repeated
ANSWER: It was corrected in the text.
Line 340: change ‘Caenorhabditis elegans (C. elegans)’ to ‘C. elegans’
ANSWER: It was corrected in the text.
Line 367: change/remove ‘X.X.’
ANSWER: It was corrected in the text.
Other questions/comments:
In Table 1, the overexpression of the genes indicated encode for what? I think you should include this in the table or in the discussion. Eg. MDR1 encodes for MFS transporters. This will clarify the choice of Candida
ANSWER: This information was included in Table 1 and in Material and Methods where the strains are described (topic 5.1).
Line 84: It is said that ‘clinical isolates are highly resistant to fluconazole (MIC > 256 μg/ml) and present different resistance mechanisms as pointed out in Material and Methods section.’ These resistance mechanisms are not pointed out in the Material and Methods section.
ANSWER: Actually, the description of those strains, showing which genes are expressed by them, explain the mechanisms involved in resistance to azoles for each one. But, to better comprehension, we decided to include more information about it at Material and Methods
In Table 2, is it possible to give the exact MIC values for Fluconazole for the control strains (instead of reporting <8)?
ANSWER: Since values < 8.0 ug/ml are considered sensible, using the CLSI standard, we started our concentration curve for FLC with 8.0 ug/ml of azole, so, if there is no growth at this concentration, we considered the strain sensible and used “< 8.0 ug/ml” in the table 2. We thought that was ok represents like that but, if you believe that is necessary to put the right value for each ATCC strain, in relation to FLC, we can do another curve with values below 8.0 ug/ml of FLC, but we were afraid about deadline limited by Journal.
Is it possible to combine Figures 1 & 2 since the captions are nearly identical?
ANSWER: Figures 1 and 2 were merged in one (new Figure 1).
In Table 3, label the first line as ‘Candida strains’. The middle half of this table is difficult to interpret at first. Possibly change to something like follows:
|
|
|
MIC combined (μg/ml) |
|
|
Aureo/Fluco |
0.0156/32 |
0.03125/16 |
0.015/128 |
|
Myr/Fluco |
0.0624/64 |
0.25/32 |
0.0625/16 |
ANSWER: Table 3 was modified as suggested to make it easier to interpret.
Please check the FICI values reports in table 3. When using the equation outlined in the Material and Methods section, different values are obtained. In particular, for Candida strain 12-99 the FICI value for Aureo/Fluco should be greater than 0.5 indicating no synergic interaction.
ANSWER: FICI was revised and new values were added in the text.
Line 167: Include the amount/percentage of DMSO present in the control.
ANSWER: It was added in the text.
Line 173: Myriocin is found to be partially toxic. How will this effect it’s potential as a therapeutic agent for Candida infections?
ANSWER: Considering the cytotoxicity assay, myrocin was partially toxic at 0.313 µg/ml, but macrophages kept around 75% of their viability. This concentration is 5-fold higher than the MIC found when myriocin is combined with fluconazole. When we analyze the C. elegans assay, myriocin toxicity was even lower, because C. elegans survival was higher than 95% in the presence of 0.5 µg/ml, which represents 8-fold higher compared to the MIC combined with fluconazole. Thus, we believe that myriocin toxicity would not impair its use as a synergistic approach.
Figure 5: Please include the line for the control from 0-1 day. It is unclear whether it is at 99 or 100 %.
ANSWER: In fact, both values (relative numbers of live worms) for control and aureobasidin are the same so, the lines are overlapping, but to make the view clear we decided to put the control line (black line) over the aureobasidin line (green line) that became different from control after one day of experiment.
Figure 5: I suggest using a different colour line for the control in the presence of 5% DMSO in the inset. This will clearly differentiate between the two controls in this figure.
ANSWER: Actually, the point did in the presence of 5% of DMSO was just to prove that is possible to kill the worms with a simple solution but, it was possible to check that after 4 days of experiments, so, to make clear and for better understanding, we decided to take out the control presented at graph inset.
Lines 210-212: This reference and example is discussed again in lines 272-276. It is not necessary to mention twice in the discussion.
ANSWER: It was corrected in the text.
Line 309-310: The XTT reduction assay is mentioned here. It would be good to include a brief summary of the procedure you used.
ANSWER: XTT procedure was added to the text.
Reviewer 3 Report
The study assessed the activity of two sphingolipid synthesis inhibitors, aureobasidin A and myriocin, with and withot fluconazole, against Candida species, as well as the potential cytotoxic effects. The results and methodology need some improvements for a more clear vision:
Methods:
chap 4.2 - please shortly describe the MIC methodology and XTT assay
chap. 4.3 - the paragraphs are doubled.
Results:
line 83 - the resistance mechanisms are not presented in materials and methods (or Table 1 should be moved there).
chap. 2.1 - was also the MIC for fluconazole assessed in this experiment? Table 2 presents <8 ug/ml; should be <1 for susceptible Candida strains.
Figure 2 - the legend should include also the Candida names, not only the strain number
Table 3 - not clear what Aureo + Fluco or Myr + Fluco very small values are, compared to the in-text presented data (Fluco + Aureo, Fluco + Myr)
Figure 5 - what is the purpose of the 5% DMSO survival curve? This concentration was not described in the methods.
English grammar and punctuation needs rechecking.
Author Response
REVIEWER 3:
Methods:
chap 4.2 - please shortly describe the MIC methodology and XTT assay
ANSWER: A better MIC and XTT methodology was added to the text.
chap. 4.3 - the paragraphs are doubled.
ANSWER: The doubled paragraph was removed.
Results:
line 83 - the resistance mechanisms are not presented in materials and methods (or Table 1 should be moved there).
ANSWER: This information was included in Table 1 and in Material and Methods (topic 5.1).
chap. 2.1 - was also the MIC for fluconazole assessed in this experiment?
ANSWER: Fluconazole MIC values are shown in Table 2. It was classified as < 8 µg/ml or > 256 µg/ml due to the Clinical Laboratory Standards Institute (CLSI) M60 criteria to classify Candida strains as sensitive or resistant to fluconazole.
Table 2 presents <8 ug/ml; should be <1 for susceptible Candida strains.
ANSWER: As we answered to referee 2, since values < 8.0 ug/ml are considered sensible, using the CLSI standard, we started our concentration curve for FLC with 8.0 ug/ml of azole, so, if there is no growth at this concentration, we considered the strain sensible and used “< 8.0 ug/ml” in the table 2. We thought that was ok represents like that but, if you believe that is necessary to put the right value for each ATCC strain, in relation to FLC, we can do another curve, but we were afraid about deadline limited by Journal.
Figure 2 - the legend should include also the Candida names, not only the strain number
ANSWER: Since Figures 1 and 2 were combined in one figure (new Figure 1), the legend was modified, and the names were included.
Table 3 - not clear what Aureo + Fluco or Myr + Fluco very small values are, compared to the in-text presented data (Fluco + Aureo, Fluco + Myr)
ANSWER: Table 3 was modified to clarify this information.
Figure 5 - what is the purpose of the 5% DMSO survival curve? This concentration was not described in the methods.
ANSWER: Actually, the point did in the presence of 5% of DMSO was just to prove that is possible to kill the worms, but it was possible to check that after 4 days of experiments, so, to make clear and for better understanding, and since the Referee 2 pointed the same, we decided to take out the control presented at graph inset.
English grammar and punctuation need rechecking.
ANSWER: The whole text was again carefully revised to improve the English quality.
This manuscript is a resubmission of an earlier submission. The following is a list of the peer review reports and author responses from that submission.